# Provable Certificates for Adversarial Examples: Fitting a Ball in the Union of Polytopes

**Matt Jordan**[*]
University of Texas at Austin
mjordan@cs.utexas.edu

**Justin Lewis**[*]
University of Texas at Austin
justin94lewis@utexas.edu

**Alexandros G. Dimakis**
University of Texas at Austin
dimakis@austin.utexas.edu

## Abstract

We propose a novel method for computing exact pointwise robustness of deep neural networks for all convex $\ell_p$ norms. Our algorithm, GeoCert, finds the largest $\ell_p$ ball centered at an input point $x_0$, within which the output class of a given neural network with ReLU nonlinearities remains unchanged. We relate the problem of computing pointwise robustness of these networks to that of computing the maximum norm ball with a fixed center that can be contained in a non-convex polytope. This is a challenging problem in general, however we show that there exists an efficient algorithm to compute this for polyhedral complices. Further we show that piecewise linear neural networks partition the input space into a polyhedral complex. Our algorithm has the ability to almost immediately output a nontrivial lower bound to the pointwise robustness which is iteratively improved until it ultimately becomes tight. We empirically show that our approach generates distance lower bounds that are tighter compared to prior work, under moderate time constraints.

## 1 Introduction

The problem we consider in this paper is that of finding the $\ell_p$-pointwise robustness of a neural net with ReLU nonlinearities with respect to general $\ell_p$ norms. The pointwise robustness of a neural net classifier, $f$, for a given input point $x_0$ is defined as the smallest distance from $x_0$ to the decision boundary [1]. Formally, this is defined as

$$\rho(f, x_0, p) := \inf_x \{\epsilon \geq 0 \mid f(x) \neq f(x_0) \ \wedge \ ||x - x_0||_p = \epsilon\}. \tag{1}$$

Computing the pointwise robustness is the central problem in certifying that neural nets are robust to adversarial attacks. Exactly computing this quantity this problem has been shown to be NP-complete in the $\ell_\infty$ setting [11], with hardness of approximation results under the $\ell_1$ norm [25]. Despite these hardness results, multiple algorithms have been devised to exactly compute the pointwise robustness, though they may require exponential time in the worst case. As a result, efficient algorithms have also been developed to give provable lower bounds to the pointwise robustness, though these lower bounds may be quite loose.

In this work, we propose an algorithm that initially outputs a nontrivial lower bound to the pointwise robustness and continually improves this lower bound until it becomes tight. Although our algorithm

---

[*]First two authors have equal contribution

has performance which is theoretically poor in the worst case, we find that in practice it provides a fundamental compromise between the two extremes of complete and incomplete verifiers. This is useful in the case where a lower-bound to the pointwise robustness is desired under a moderate time budget.

The central mathematical problem we address is how to find the largest $\ell_p$ ball with a fixed center contained in the union of convex polytopes. We approach this by decomposing the boundary of such a union into convex components. This boundary may have complexity exponential in the dimension in the general case. However, if the polytopes form a polyhedral complex, an efficient boundary decomposition exists and we leverage this to develop an efficient algorithm to compute the largest $\ell_p$ ball with a fixed center contained in the polyhedral complex. We connect this geometric result to the problem of computing the pointwise robustness of piecewise linear neural networks by proving that the linear regions of piecewise linear neural networks indeed form a polyhedral complex. Further, we leverage the lipschitz continuity of neural networks to both initialize at a nontrivial lower bound, and guide our search to tighten this lower bound more quickly.

Our contributions are as follows:

- We provide results on the boundary complexity of polyhedral complices, and use these results to motivate an algorithm to compute the the largest interior $\ell_p$ ball centered at $x_0$.
- We prove that the linear regions of piecewise linear neural networks partition the input space into a polyhedral complex.
- We incorporate existing incomplete verifiers to improve our algorithm and demonstrate that under a moderate time budget, our approach can provide tighter lower bounds compared to prior work.

## 2   Related Work

**Complete Verifiers:**   We say that an algorithm is a *complete verifier* if it exactly computes the pointwise robustness of a neural network. Although this problem is NP-Complete in general under an $\ell_\infty$ norm [11], there are two main algorithms to do so. The first leverages formal logic and SMT solvers to generate a certificate of robustness [11], though this approach only works for $\ell_\infty$ norms. The second formulates certification of piecewise linear neural networks as mixed integer programs and relies on fast MIP solvers to be scalable to reasonably small networks trained on MNIST [20, 8, 13, 6, 4]. This approach extends to the $\ell_2$ domain so long as the mixed integer programming solver utilized can solve linearly-constrained quadratic programs [20]. Both of these approaches are fundamentally different than our proposed method and do not provide a sequence of ever-tightening lower bounds. Certainly each can be used to certify any given lower bound, or provide a counterexample, but the standard technique to do so is unable to reuse previous computation.

**Incomplete Verifiers:**   There has been a large body of work on algorithms that output a certifiable lower bound on the pointwise robustness. We call these techniques *incomplete verifiers*. These approaches employ a variety of relaxation techniques. Linear programming approaches admit efficient convex relaxations that can provide nontrivial lower bounds [26, 25, 17, 7]. Exactly computing the Lipschitz constant of neural networks has also been shown to be NP-hard [22], but overestimations of the Lipschitz constant have been shown to provide lower bounds to the pointwise robustness [15, 25, 19, 10, 21]. Other relaxations, such as those leveraging semidefinite programming, or abstract representations with zonotopes are also able to provide provable lower bounds [16, 14]. An equivalent formulation of this problem is providing overestimations on the range of neural nets, for which interval arithmetic has been shown useful [23, 24]. Other approaches generate lower bounds by examining only a single linear region of a PLNN [18, 5], though we extend these results to arbitrarily many linear regions. These approaches, while typically more efficient, may provide loose lower bounds.

## 3   Centered Chebyshev Ball

**Notations and Assumptions**
Before we proceed, we introduce some notation. A *convex polytope* is a bounded subset of $\mathbb{R}^n$ that

can be described as the intersection of a finite number of halfspaces. The polytopes we study are described succinctly by their linear inequalities (i.e., they are H-polytopes), which means that the number of halfspaces defining the polytope, denoted by $m$, is at most $\mathcal{O}(poly(n))$, i.e. polynomial in the ambient dimension. If a polytope $\mathcal{P}$ is described as $\{x \mid Ax \leq b\}$, an $(n-k)$-*face* of $\mathcal{P}$ is a nonempty subset of $\mathcal{P}$ defined as the set $\{x \mid x \in \mathcal{P} \ \wedge \ A^= x = b^=\}$ where $A^=$ is a matrix of rank $k$ composed of a subset of the rows of $A$, and $b^=$ is the corresponding subset of $b$. We use the term *facet* to refer to an $(n-1)$ face of $\mathcal{P}$. We define the boundary $\delta \mathcal{P}$ of a polytope as the union of the facets of $\mathcal{P}$. We use the term *nonconvex polytope* to describe a subset of $\mathbb{R}^n$ that can be written as a union of finitely many convex polytopes, each with nonempty interior. The $\ell_p$-norm ball of size $t$ centered at point $x_0$ is denoted by $B_t^p(x_0) := \{x \mid ||x - x_0||_p \leq t\}$. The results presented hold for $\ell_p$ norms for $p \geq 1$. When the choice of norm is arbitrary, we use $|| \cdot ||$ to denote the norm and $B_t(x_0)$ to refer to the corresponding norm ball.

**Centered Chebyshev Balls:** Working towards the case of a union of polytopes, we first consider the simple case of fitting the largest $\ell_p$-ball with a fixed center inside a single polytope. The uncentered version of this problem is typically referred to as finding the *Chebyshev center* of a polytope and can be computed via a single linear program [3, 2]. When the center is fixed, this can be viewed as computing the projection to the boundary of the polytope. In fact, in the case for a single polytope, it suffices to compute the projection onto the hyperplanes containing each facet. See Appendix A for further discussion computing projections onto polytopes. Ultimately, because of the polytope's geometric structure, the problem's decomposition is straightforward. This theme of efficient boundary decomposition will prove to hold true for polyhedral complices as shown in the following sections.

Now, we turn our attention to the case of finding a centered Chebyshev ball inside a general nonconvex polytope. This amounts to computing the projection to the boundary of the region. The key idea here is that the boundary of a nonconvex polytope can be described as the union of finitely many $(n-1)$-dimensional polytopes; however, the decomposition may be quite complex. We define this set formally as follows:

**Definition 1.** *The **boundary** of a non-convex polytope $P$ is the largest set $T \subseteq P$ such that every point $x \in T$ satisfies the following two properties:*

  *(i) There exists an $\epsilon_0$ and a direction $u$ such that for all $\epsilon \in (0, \epsilon_0)$, there exists a neighborhood centered around $x + \epsilon u$ that is contained in $P$.*

  *(ii) There exists an $\eta_0$ and a direction $v$ such that for all $\eta \in (0, \eta_0)$, $x + \eta v \notin P$.*

The boundary is composed of finitely many convex polytopes, and computing the projection to a single convex polytope is an efficiently computable convex program. If there exists an efficient decomposition of the boundary of a nonconvex polytope into convex sets, then a viable algorithm is to simply compute the minimal distance from $x_0$ to each component of the boundary and return the minimum. Unfortunately, for general nonconvex polytopes, there may not be an efficient convex decomposition. See Theorem B.1 in Appendix B.

However, there do exist classes of nonconvex polytopes that admit a convex decomposition with size that is no larger than the description of the nonconvex polytope itself. To this end, we introduce the following definition (see also Ch. 5 of [27]):

**Definition 2.** *A nonconvex polytope, described as the union of elements of the set $\mathscr{P} = \{\mathcal{P}_1, ..., \mathcal{P}_k\}$ forms a **polyhedral complex** if, for every $\mathcal{P}_i, \mathcal{P}_j \in \mathscr{P}$ with nonempty intersection, $\mathcal{P}_i \cap \mathcal{P}_j$ is a face of both $\mathcal{P}_i$ and $\mathcal{P}_j$. Additionally, for brevity, if a pair of polytopes $\mathcal{P}, \mathcal{Q}$, form a polyhedral complex, we say they are **PC**. (See Figure 2 for examples.)*

We can now state our main theorem concerning the computation of the centered Chebyshev ball within polyhedral complices:

**Theorem 3.1.** *Given a polyhedral complex, $\mathscr{P} = \{\mathcal{P}_1, \dots \mathcal{P}_k\}$, where $\mathcal{P}_i$ is defined as the intersection of $m_i$ closed halfspaces. Let $M = \sum_i m_i$, and let $x_0$ be a point contained by at least one such $\mathcal{P}_i$. Then the boundary of $\bigcup_{i \in [k]} \mathcal{P}_i$ is represented by at most $M$ $(n-1)$-dimensional polytopes. There exists an algorithm that can compute this boundary in $\mathcal{O}(poly(n, M, k))$ time.*

Returning to our desired application, we now prove a corollary about the centered Chebyshev ball contained in a union of polytopes.

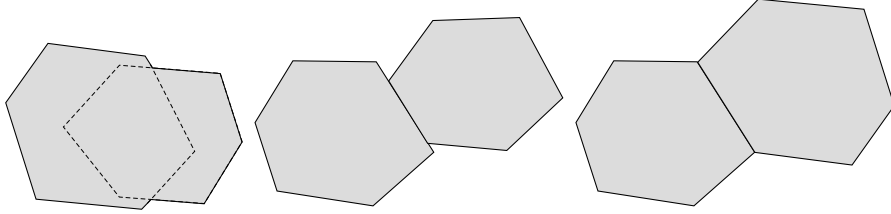

Figure 1: Three potential configurations of a nonconvex polytope. Note that only the rightmost nonconvex polytope forms a polyhedral complex.

**Corollary 3.2.** *Given a collection, $\mathscr{P} = \{\mathcal{P}_1, \ldots \mathcal{P}_k\}$ that meets all the conditions outlined in theorem 3.1, with the boundary of $\mathscr{P}$ computed as in theorem 3.1, the centered Chebyshev ball around $x_0$ has size*

$$t := \inf_{x \in T} ||x - x_0|| \tag{2}$$

*This can be solved by at most $M$ linear programs in the case of $\ell_\infty$ norm, or at most $M$ linearly constrained quadratic programs in the case of the $\ell_2$-norm.*

**Graph Theoretic Formulation:**

Theorem 3.1 and its corollary provide a natural algorithm to computing the centered Chebyshev ball of a polyhedral complex: compute the convex components of the boundary and then compute the projection to each component. In the desired application of computing robustness of neural networks, the number of such convex components may be large and therefore it may be inefficient to even enumerate each component. While we demonstrate in Appendix G that the number of linear regions of ReLU networks tends to be much smaller than their theoretical upper bound, it is of interest to develop algorithms that do not have to compute projections to *every* boundary facet. In the absence of other information, one must at least compute the projection to every facet, boundary or otherwise, intersecting the centered Chebyshev ball.

A more natural way to view this problem is as a local-search problem along a bipartite graph. For a given polyhedral complex $\mathscr{P}$ composed of polytopes $\mathcal{P}_1, \mathcal{P}_2, \ldots$, we construct a bipartite graph where each right vertex correspond to an $n$-dimensional polytope $\mathcal{P}_i$, and each left vertex corresponds to a facet of the polyhedral complex. We abuse notation and let $\mathcal{P}_i$ refer to the right-vertex and its corresponding polytope and similarly for left-vertex and facet $\mathcal{F}_j$. An edge exists between right-vertex $\mathcal{P}_i$ and left-vertex $\mathcal{F}_j$ iff polytope $\mathcal{P}_i$ contains facet $\mathcal{F}_j$. In other words, the graph of interest is composed of the terminal elements of the face lattice and their direct ancestors. By definition, for any polyhedral complex, the left-degree of this graph is at most 2.

In the context of computing the centered Chebyshev ball, centered around a point $x_0$, we further equip each left-vertex/facet $\mathcal{F}_j$ in our graph with a value which we refer to as the 'potential.' For now, the potential of vertex $\mathcal{F}_j$ can be thought of as the projection distance between $x_0$ and the facet $\mathcal{F}_j$. We will denote the potential of vertex $\mathcal{F}_j$ as $\Phi(\mathcal{F}_j)$. The boundary facets, $T$, correspond to a subset of the left-vertices and recall that our goal is to return the left-vertex with minimal potential. By the triangle inequality, any ray starting at $x_0$ that intersects multiple facets in order $\mathcal{F}_{i_1}, \mathcal{F}_{i_2}, \ldots$ will have that $\Phi(\mathcal{F}_{i_1}) \leq \Phi(\mathcal{F}_{i_2}) \leq \ldots$. Further, one can represent any norm ball $B_t(x_0)$ as a subset $S_t$ of left and right vertices of the graph. A left-vertex $\mathcal{F}_j$ is in $S_t$ iff $\Phi(\mathcal{F}_j) \leq t$.

The local search along this graph can be thought of as follows. Any point $x_0$ contained inside a polyhedral complex must reside in at least one polytope $\mathcal{P}_i$, and our goal is to find the facet with minimum potential. The idea is similar to Djikstra's algorithm, where we maintain a set of 'frontier facets' in a priority queue, ordered by their potential $\Phi$, and a set of right-vertices/polytopes which have already been explored. At each iteration, we pop the frontier facet with minimal potential, and examine its neighbors, which correspond to polytopes containing this facet. Since the left-degree of the graph is 2, at most one of these neighboring polytopes has not yet been explored. If such a polytope exists, for each of its neigbors/facets, we compute the potential and insert the facet into the priority queue of 'frontier facets', and also add this new polytope to our set of explored polytopes. At initialization, the set of seen polytopes is composed only of the polytope containing $x_0$, and termination occurs as soon as a boundary facet is popped from the priority queue. Pseudocode for this procedure is outlined in Algorithm 1 and a proof of correctness is provided in Appendix C.

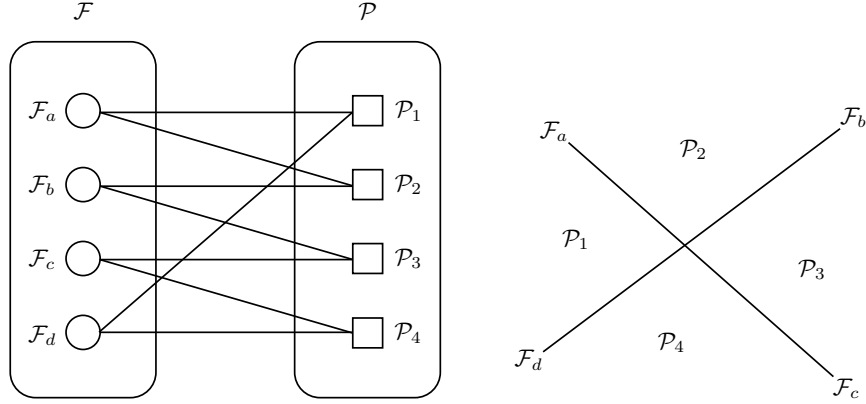

Figure 2: Example of bipartite graph defined over facets $\mathcal{F}$ and polytopes $\mathcal{P}$ of polyhedral complex $\mathscr{P}$. Note that each facet is shared by at most two polytopes.

**: Algorithm 1: GeoCert**

**Input:** point $x_0$, potential $\Phi$;
**Initialization:** ;
// Setup priority queue, seen-polytope set;
$Q \leftarrow [\ ]; C \leftarrow \{\mathcal{P}(x_0)\}$;
// Handle first polytope's facets;
**for** *Facet* $\mathcal{F} \in N(\mathcal{P}(x_0))$ **do**
    $Q.push((\Phi(\mathcal{F}), \mathcal{F}))$;
**end**
// Loop until boundary is popped;
**while** $Q \neq \emptyset$ **do**
    $\mathcal{F} \leftarrow Q.pop()$;
    **if** $\mathcal{F}$ *is boundary* **then**
        **Return** $\mathcal{F}$;
    **else**
        **for** $\mathcal{P} \in N(\mathcal{F}) \setminus C$ **do**
            **for** $\mathcal{F} \in N(P)$ **do**
                $Q.push((\Phi(\mathcal{F}), \mathcal{F}))$;
            **end**
        **end**
    **end**
**end**

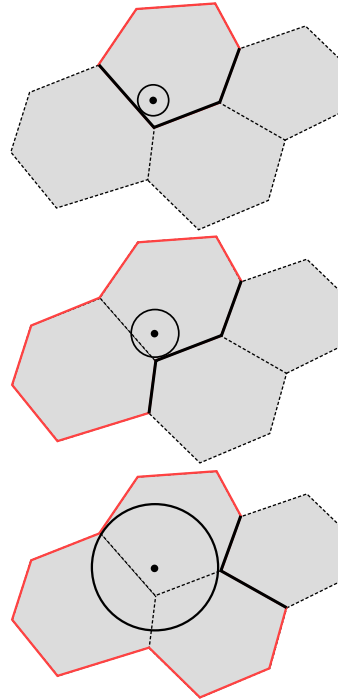

Figure 3: Pseudocode for GeoCert (left) and a pictorial representation of the algorithm's behavior on a simple example (right). The facets colored belong to the priority queue, with red and black denoting adversarial facets and non-adversarial facets respectively. Once the minimal facet in the queue is adversarial, the algorithm stops.

This alternative phrasing of our problem aids us in two ways. First, we note that the potential of any left-vertex $\mathcal{F}_j$ can be computed as needed. Indeed, letting $t^*$ be the minimum potential of all facets contained in the boundary set, this search procedure only requires that the potential need only be computed for the facets contained in $S_{t^*}$, as opposed to the entire collection of facets. Second, $\Phi$ need not refer to the euclidean projection distance, and alternative potential functions exist which further reduce the number of potentials that need to be computed while preserving correctness. These will be further discussed in Section 5 and Appendix C.

**Iteratively Constructing Polyhedral Complices**
Finally, we note an approach by which polyhedral complices may be formed that will become useful when we discuss PLNN's in the following section. We present the following three lemmas which

relate to iterative constructions of polyhedral complices. Informally, they state that given any polytope or pair of polytopes which are PC, a slice with a hyperplane or a global intersection with a polytope generates a set that is still PC.

**Lemma 3.3.** *Given an arbitrary polytope $\mathcal{P} := \{x \mid Ax \leq b\}$ and a hyperplane $\mathcal{H} := \{x \mid c^T x = d\}$ that intersects the interior of $\mathcal{P}$, the two polytopes formed by the intersection of $\mathcal{P}$ and the each of closed halfpsaces defined by $\mathcal{H}$ are PC.*

**Lemma 3.4.** *Let $\mathcal{P}, \mathcal{Q}$ be two PC polytopes and let $H_{\mathcal{P}}, H_{\mathcal{Q}}$ be two hyperplanes that define two closed halfspaces each, $H_{\mathcal{P}}^+, H_{\mathcal{P}}^-, H_{\mathcal{Q}}^+, H_{\mathcal{Q}}^-$. If $\mathcal{P} \cap \mathcal{Q} \cap H_{\mathcal{P}} = \mathcal{P} \cap \mathcal{Q} \cap H_{\mathcal{Q}}$ then the subset of the four resulting polytopes $\{\mathcal{P} \cap H_{\mathcal{P}}^+, \mathcal{P} \cap H_{\mathcal{P}}^-, \mathcal{Q} \cap H_{\mathcal{Q}}^+, \mathcal{Q} \cap H_{\mathcal{Q}}^-\}$ with nonempty interior forms a polyhedral complex.*

And the following will be necessary when we handle the case where we wish to compute the pointwise robustness for the image classification domain, where valid images are typically defined as vectors contained in the hypercube $[0, 1]^n$.

**Lemma 3.5.** *Let $\mathscr{P} = \{\mathcal{P}_1, \dots \mathcal{P}_k\}$ be a polyhedral complex and let $\mathcal{D}$ be any polytope. Then the set $\{\mathcal{P}_i \cap \mathcal{D} \mid \mathcal{P}_i \in \mathscr{P}\}$ also forms a polyhedral complex.*

# 4 Piecewise Linear Neural Networks

We now demonstrate an application of the geometric results described above to certifying robustness of neural nets. We only discuss networks with fully connected layers and ReLU nonlinearities, but our results hold for networks with convolutional and skip layers as well as max and average pooling layers. Let $f$ be an arbitrary $L$-layer feed forward neural net with fully connected layers and ReLU nonlinearities, where each layer $f^{(i)} : \mathbb{R}^{n_{i-1}} \to \mathbb{R}^{n_i}$ has the form

$$f^{(i)}(x) = \begin{cases} W_i x + b_i, & \text{if i = 1} \\ W_i \sigma(f^{(i-1)}(x)) + b_i, & \text{if } i > 1 \end{cases} \quad (3)$$

where $\sigma$ refers to the element-wise ReLU operator. And we denote the final layer output $f^{(L)}(x)$ as $f(x)$. We typically use the capital $F(x)$ to refer to the maximum index of $f$: $F(x) := \arg\max_i f_i(x)$. We define the *decision region* of $f$ at $x_0$ as the set of points for which the classifier returns the same label as it does for $x_0$: $\{x \mid F(x) = F(x_0)\}$.

It is important to note is that $f^{(i)}(x)$ refers to the *pre-ReLU* activations of the $i^{th}$ layer of $f$. Let $m$ be the number of neurons of $f$, that is $m = \sum_{i=1}^{L-1} n_i$. We describe a neuron configuration as a ternary vector, $A \in \{-1, 0, 1\}^m$, such that each coordinate of $A$ corresponds to a particular neuron in $f$. In particular, for neuron $j$,

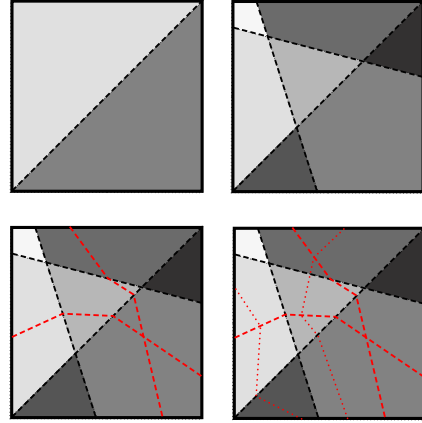

Figure 4: Pictorial reference for proof of Theorem 4.2. *(Top Left)* A single Relu activation partitions the input space into two PC polytopes *(Top Right)* as additional activations are added at the first layer, the collection is still PC by Lemma 3.4. *(Bottom Left)* as the next layer of activations are added, the partitioning is linear within each region created previously and PC at the previous boundaries, thus still PC. *(Bottom Right)* the partitioning due to all subsequent layers preserves PC-ness by induction.

$$A_j = \begin{cases} +1, & \text{if neuron } j \text{ is 'on'} \\ -1, & \text{if neuron } j \text{ is 'off'} \\ 0, & \text{if neuron } j \text{ is both 'on' and 'off'} \end{cases} \quad (4)$$

Where a neuron being 'on' corresponds to the pre-ReLU activation is at least zero, 'off' corresponds to the pre-ReLU being at most zero, and if a neuron is both on and off its pre-ReLU activation is identically zero. Further each neuron configuration corresponds to a set

$$\mathcal{P}_A = \{x \mid f(x) \text{ has neuron activation consistent with } A\}$$

The following have been proved before, but we include them to introduce notational familiarity:

**Lemma 4.1.** *For a given neuron configuration $A$, the following are true about $\mathcal{P}_A$,*

*(i) $f^{(i)}(x)$ is linear in $x$ for all $x \in \mathcal{P}_A$.*

*(ii) $\mathcal{P}_A$ is a polytope.*

This lets us connect the polyhedral complex results from the previous section towards computing the pointwise robustness of PLNNs. Letting the potential $\phi$ be the $\ell_p$ distance, we can apply Algorithm 1 towards this problem.

**Theorem 4.2.** *The collection of $\mathcal{P}_A$ for all $A$, such that $\mathcal{P}_A$ has nonempty interior forms a polyhedral complex. Further, the decision region of $F$ at $x_0$ also forms a polyhedral complex.*

In fact, except for a set of measure zero over the parameter space, the facets of each such linear region correspond to exactly one ReLU flipping configurations:

**Corollary 4.3.** *If the network parameters are in general position and $A, B$ are neuron configurations such that $dim(\mathcal{P}_A) = dim(\mathcal{P}_B) = n$ and their intersection is of dimension $(n-1)$, then $A, B$ have hamming distance 1 and their intersection corresponds to exactly one ReLU flipping signs.*

## 5   Speedups

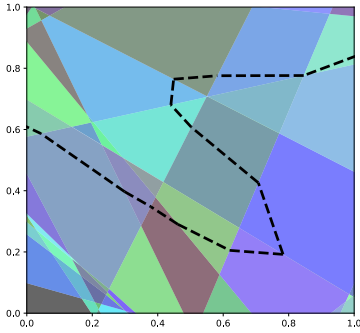

Figure 5: Piecewise Linear Regions of a 2D toy network. The dotted line represents the decision boundary.

While our results in section 3 hold for general polyhedral complices, we can boost the performance of GeoCert by leveraging additional structure of PLNNs. As the runtime of GeoCert hinges upon the total number of iterations and time per iteration, we discuss techniques to improve each.

**Improving Iteration Speed Via Upper Bounds**
At each iteration, GeoCert pops the minimal element from the priority queue of 'frontier facets' and, using the graph theoretic lens, considers the facets in its two-hop neighborhood. Geometrically this corresponds to popping the minimal-distance facet seen so far, considering the polytope on the opposite side of that facet and computing the distan1ce to each of its facets. In the worst case, the number of facets of each linear region is the number of ReLU's in the PLNN. While computing the projection requires a linear or quadratic program, as we will show, it is usually not necessary to compute a convex program for each every nonlinearity at every iteration.

If we can quickly guarantee that a potential facet is infeasible within the domain of interest then we avoid computing the projection exactly. In the image classification domain, the domain of valid images is usually the unit hypercube. If an upper bound on the pointwise robustness, $U$, is known, then it suffices to restrict our domain to $\mathcal{D}' := B_U(x_0) \cap \mathcal{D}$. This aids us in two ways: (i) if the hyperplane containing a facet does not intersect $\mathcal{D}'$ then the facet also does not intersect $\mathcal{D}'$; (ii) a tighter restriction on the domain allows for tighter bounds on pre-ReLU activations. For point (i), we observe that computing the feasibility of the intersection of a hyperplane and hyperbox is linear in the dimension and hence many facets can very quickly be deemed infeasible. For point (ii), if we can guarantee ReLU stability, by Corollary 4.3, then we can deem the facets corresponding to the each stable ReLU as infeasible. ReLU stability additionally provides tighter upper bounds on the Lipschitz constants of the network.

Any valid adversarial example provides an upper bound on the pointwise robustness. Any point on any facet on the boundary of the decision region also provides an upper bound. In Appendix F, we describe a novel tweak that can be used to generate adversarial examples tailored to be close to the original point. Also, during the runtime of Geocert, any time a boundary facet is added to the priority queue, we update the upper bound based on the projection magnitude to this facet.

**Improving Number of Iterations Via Lipschitz Overestimation**
When one uses distance as a potential function, if the true pointwise robustness is $\rho$, then GeoCert

must examine every polytope that intersects $B_\rho(x_0)$. This is necessary in the case when no extra information is known about the polyhedral complex of interest. However one can incorporate the lipschitz-continuity of a PLNN into the potential function $\phi$ to reduce on the number of linear regions examined. The main idea is that as the network has some smoothness properties, any facet for which the classifier is very confident in its answer must be very far from the decision boundary.

**Theorem 5.1.** *Letting $F(x_0) = i$, and $g_j(x) = f_i(x) - f_j(x)$ and an upper bound $L_j$ on the lipschitz continuity of $g_j$, using $\phi_{lip}(y) := ||x_0 - y|| + \min_{j \neq i} \frac{g_j(y)}{L_j}$ as a potential for GeoCert maintains its correctness in computing the pointwise robustness.*

The intuition behind this choice of potential is that it biases the set of seen polytopes to not expand too much in directions for which the distance to the decision boundary is guaranteed to be large. This effectively is able to reduce the number of polytopes examined, and hence the number of iterations of GeoCert, while still maintaining complete verification. A critical bonus of this approach is that it allows one to 'warm-start' GeoCert with a nontrivial lower bound that will only increase until becoming tight at termination. A more thorough discussion on upper-bounding the lipschitz constant of each $g_j$ can be found in [25].

## 6 Experiments

**Exactly Computing the Pointwise Robustness:** Our first experiment compares the average pointwise robustness bounds provided by two complete verification methods, GeoCert and MIP, as well as an incomplete verifier, Fast-Lip. The average $\ell_p$ distance returned by each method and the average required time (in seconds) to achieve this bound are provided in Table 1. Verification for $\ell_2$ and $\ell_\infty$ robustness was conducted for 1000 random validation images for two networks trained on MNIST. Networks are divided into binary and non-binary examples. Binary networks were trained to distinguish a subset of 1's and 7's from the full MNIST dataset. All networks were trained with $\ell_1$ weight regularization with $\lambda$ set to $2 \times 10^{-3}$. All networks are composed of fully connected layers with ReLU activations. The layer-sizes for the two networks are as follows: i) [784, 10, 50, 10, 2] termed 70NetBin and ii) [784, 20, 20, 2] termed 40NetBin. Mixed integer programs and linear programs were solved using Gurobi [9]. The code for reproducing experiments has been made publicly available[†].

From Table 1, it is clear that Geocert and MIP return the exact robustness value while Fast-Lip provides a lower bound. While the runtimes for MIP are faster than those for GeoCert, they are within an order of magnitude. In these experiments, we record the timing when each method is left to run to completion; however, in the experiment to follow we demonstrate that GeoCert provides a non-trivial lower bound faster than other methods.

Table 1: (Left) Times (seconds) to compute exact pointwise robustness on binary MNIST networks for both the $\ell_2$ and $\ell_\infty$ settings over 1000 random examples. Boldface corresponds to the exact pointwise robustness. (Right) Provable lower bounds for a binary MNIST network under a fixed 300s time limit. Note that GeoCert initializes at the bound provided by Fast-Lip and continually improves. Boldface here corresponds to the tightest lower bound found. Note that our algorithm outperforms all previous methods for this task.

| Method | $\ell_p$ | 70NetBin Dist. | 70NetBin Time | 40NetBin Dist. | 40NetBin Time |
|---|---|---|---|---|---|
| Fast-Lip | | 0.092 | 0.012 | 0.116 | 0.009 |
| GeoCert | $\ell_\infty$ | **0.175** | 1.453 | **0.190** | 4.924 |
| MIP | | **0.175** | 0.771 | **0.190** | 0.797 |
| Fast-Lip | | 0.905 | 0.007 | 1.124 | 0.008 |
| GeoCert | $\ell_2$ | **1.414** | 2.816 | **1.533** | 6.958 |
| MIP | | **1.414** | 1.972 | **1.533** | 4.466 |

| Ex. | Fast-Lip | GeoCert | MIP |
|---|---|---|---|
| 1 | 1.782 | **2.251** | 2.0 |
| 2 | 1.319 | **1.356** | 1.0 |
| 3 | 1.501 | **1.620** | 1.0 |
| 4 | 1.975 | **2.499** | 2.0 |
| 5 | 1.871 | **2.402** | 2.0 |

**Best Lower Bound Under a Time Limit:** To demonstrate the ability of GeoCert to provide a lower bound greater than those generated by incomplete verifiers and other complete verifiers under

---

[†]https://github.com/revbucket/geometric-certificates

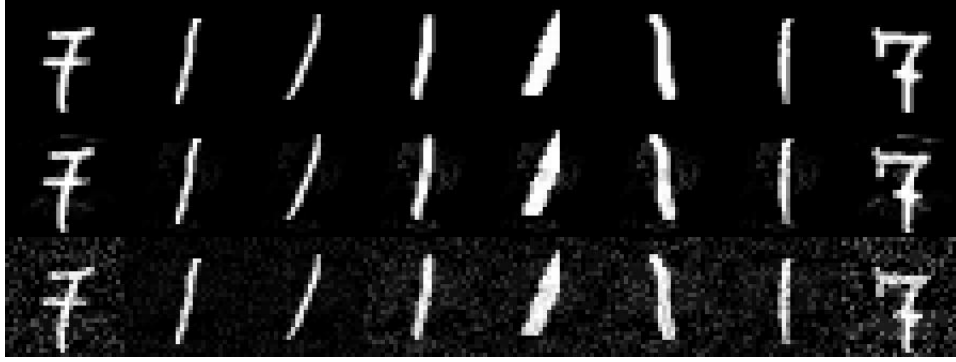

Figure 6: Original MNIST images (top) compared to their minimal distance adversarial examples as found by GeoCert (middle) and the minimal distortion adversarial attacks found by Carlini-Wagner $\ell_2$ attack. The average $\ell_2$ distortion found by GeoCert is 31.6% less that found by Carlini-Wagner.

a fixed time limit we run the following experiment. On the binary MNIST dataset, we train a network with layer sizes [784, 20, 20, 20, 2] using Adam and a weight decay of 0.02 [12]. We allow a time limit of 5 minutes per example, which is not sufficient for either GeoCert or MIP to complete. As per the codebase associated with [20], for MIP we use a binary search procedure of $\epsilon = [0.5, 1.0, 2.0, 4.0, \ldots]$ to verify increasingly larger lower bounds. We also compare against the lower bounds generated by Fast-Lip [25], noting that using the Lipschitz potential described in Section 5 allows GeoCert to immediately initialize to the bound produced by Fast-Lip. We find that in all examples considered, after 5 minutes, GeoCert is able to generate larger lower-bounds compared to MIP. Table 1 demonstrates these results for 5 randomly chosen examples.

# 7 Conclusion

This paper presents a novel approach towards both bounding and exactly computing the pointwise robustness of piecewise linear neural networks for all convex $\ell_p$ norms. Our technique differs fundamentally from existing complete verifiers in that it leverages local geometric information to continually tighten a provable lower bound. Our technique is built upon the notion of computing the centered Chebyshev ball inside a polyhedral complex. We demonstrate that polyhedral complices have efficient boundary decompositions and that each decision region of a piecewise linear neural network forms such a polyhedral complex. We leverage the Lipschitz continuity of PLNN's to immediately output a nontrivial lower bound to the pointwise robustness and improve this lower bound until it ultimately becomes tight.

We observe that mixed integer programming approaches are typically faster in computing the exact pointwise robustness compared to our method. However, our method provides intermediate valid lower bounds that are produced significantly faster. Hence, under a time constraint, our approach is able to produce distance lower bounds that are typically tighter compared to incomplete verifiers and faster compared to MIP solvers. An important direction for future work would be to optimize our implementation so that we can scale our method to larger networks. This is a critical challenge for all machine learning verification methods.

# 8 Acknowledgements

This research has been supported by NSF Grants 1618689, DMS 1723052, CCF 1763702, AF 1901292 and research gifts by Google, Western Digital and NVIDIA.

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
