[Supplementary Material]

# Supplementary Material For Provable Certificates for Adversarial Examples: Fitting a Ball in the Union of Polytopes

## A  Further discussion on Centered Chebyshev Balls

### A.1  Centered Chebyshev Ball of a Single Polytope

Here we present a more thorough discussion of the case of computing a centered Chebyshev ball for a single polytope, as well as general formulations for projections onto polytopes under various $\ell_p$ norms.

Consider a polytope $\mathcal{P} := \{x \mid Ax \leq b\}$. The problem of finding the the centered Chebyshev ball under an $\ell_p$ norm can written as the following optimization problem:

$$\max \quad t \tag{1}$$
$$\text{s.t.} \quad \sup_{||v|| \leq 1} a_i^T(x_0 + tv) \leq b_i \quad \forall i \in [m].$$

As a brief aside, note that if the center $x_0$ is not fixed, it is introduced as a variable in the optimization, and in general this requires a linear program to be solved. With a fixed center, each constraint can be rewritten as $t||a_i||_* \leq b_i - a_i^T x_0$, for $||\cdot||_*$ being the dual norm of $||\cdot||$. Thus the program becomes

$$\max \quad t \tag{2}$$
$$\text{s.t.} \quad t \leq \frac{b_i - a_i^T x_0}{||a_i||_*} \quad \forall i \in [m]$$

which can be solved as taking the minimum over all $i$ of $\frac{b_i - a_i^T x_0}{||a_i||_*}$. Understanding what is occurring here will be central to our theorems, so we decompose the above problem. Note that each constraint $a_i^T x \leq b_i$ defines a hyperplane, and $\frac{b_i - a_i^T x_0}{||a_i||_*}$ denotes the $\ell_p$ distance from $x_0$ to that hyperplane. In other words, this provides a lower bound on the $\ell_p$ distance to the facet of $\mathcal{P}$ generated by constraint $i$ being tight. However, the minimum of these lower bounds must be tight for the constraint that bounds the centered Chebyshev ball and therefore it suffices to compute this lower bound everywhere. Finding the centered Chebyshev ball is equivalent to finding the minimum distance to each component of the boundary of $\mathcal{P}$. An alternative, albeit more laborious, solution to finding the centered Chebyshev ball is to consider the minimal $\ell_p$ distance to $\delta\mathcal{P}$ directly by computing the $\ell_p$ distance to each facet of $\mathcal{P}$ and taking the minimum.

Figure 1: Pictorial examples of computing the centered Chebyshev ball for the $\ell_2, \ell_\infty$ norms.

## A.2  Projections onto Polytopes

As our algorithm heavily relies on the ability to efficiently compute the projection to a facet, which is itself a polytope, we describe the general formulation here. Formally, provided a polytope $\mathcal{P} := \{x \mid Ax \leq b\}$ and a point $x_0 \notin \mathcal{P}$, we wish to compute $\min_{x \in \mathcal{P}} ||x_0 - x||_p$. To compute this exactly, we decompose $x$ in the minimum to $x_0 + v$ and optimize over $v$. This is a linear program in the $\ell_1$ case, and a linearly constrained quadratic program in the $\ell_2$ case. For $\ell_\infty$ we introduce $n + 1$ auxiliary variables and $2n$ additional constraints:

$$\min_{t,v} \quad t \tag{3}$$
$$\text{s.t.} \quad A(x_0 + v) \leq b$$
$$t \geq 0$$
$$-t \cdot \mathbb{1} \leq v \leq t \cdot \mathbb{1}$$
$$\tag{4}$$

In the $\ell_1$ case, we require $2n$ auxiliary variables:

$$\min_{t,v} \quad \sum t_i \tag{5}$$
$$\text{s.t.} \quad A(x_0 + v) \leq b$$
$$t \geq 0$$
$$-t_i \leq v_i \leq t_i \quad \forall i \in [n]$$
$$\tag{6}$$

And in the case of the $\ell_2$-norm, the objective becomes quadratic while the constraints remain linear:

$$\min_{v} \quad \sum_i v_i^2 \tag{7}$$
$$\text{s.t.} \quad A(x_0 + v) \leq b$$

In both cases there exist polynomial time algorithms to solve these exactly and efficient implementations to solve these quickly in practice [2, 9]. Thus, we can solve the problem of finding the *centered Chebyshev ball* of a single polytope by solving the minimum distance to each facet, each formulated as an efficient LP or QP.

## A.3  Notes on Hyperplanes

Additionally we mention some cheap tricks that are useful when the polytopes of interest are $(n-1)$-dimensional. This implies that they lie entirely in some $(n-1)$-dimensional affine subspace, say

Figure 2: Pictorial aid for Theorem B.1

$\mathcal{P} \subseteq H$ for $H := \{x \mid a^T x = b\}$. To compute a lower-bound on the projection of $x_0$ onto $\mathcal{P}$, one can compute the projection of $x_0$ onto $H$, which can be done in linear time in the dimension:

$$\min_{t,v} \quad t \tag{8}$$
$$\text{s.t.} \quad a^t(x_0 + v) \leq b$$
$$||v|| = 1$$

Reformulating the first constraint, one has $t = \frac{b - a^T x_0}{a^T v}$. This quantity is minimzed when $a^T v$ is maximized, and $\max_{||v||=1} a^T v$ is, by definition, the dual norm $|| \cdot ||_*$ of $a$. Hence the projection onto a hyperplane is $\frac{b - a^T x_0}{||a||_*}$.

In section 5, we mention that it is efficient to compute the feasibility of $H \cap B$ for $B$ being some hyperbox defined by coordinate lower and upper bound vectors, $l$ and $u$ as $\{x \mid l \leq x \leq u\}$. We can decompose $a$ into its nonnegative components $a^+$ and its negative components $a^-$ such that $H = \{x \mid (a^+ + a^-)^T x = b\}$. Then, by interval arithmetic, we notice that the set $\{c \mid a^T x \quad \forall x \in B\}$ is the interval $[(a^+)^T l + (a^-)^T u, (a^-)^T l + (a^+)^T u]$. Iff $b$ is contained in this interval, then the intersection $H \cap B$ is nonempty.

## B   Proofs about Boundary Decompositions

Here we prove our theorems about efficient boundary decomposotions of polyhedral complices. First we state a hardness result that claims that for arbitrary nonconvex polytopes, the size of the smallest convex decomposition of the boundary may be exponential in the dimension.

**Theorem B.1.** *There exists a collection of polytopes $\mathscr{P} = \{\mathcal{P}_1, \ldots \mathcal{P}_k\}$ each with dimension $n$ and 2 constraints (for a total of $2k$ constraints) such that the boundary of $\bigcup_{i \in [k]} \mathcal{P}_i$ is composed of $\Omega(k^{n-1})$ convex components.*

*Proof.* We prove this by construction. We rely crucially on a result from hyperplane arrangements. It is a classical result that given a choice in placement of $m$ hyperplanes in $\mathbb{R}^n$, the maximum number of regions that can be generated is given, in closed form as $R(n, m) := 1 + \sum_{j=1}^{n} \binom{m}{j}$ [6]. Leveraging this, we construct our polytopes. Let $\mathcal{P}_1 = \{x \mid 0 \leq x_1 \leq 1\}$ such that it has exactly two facets, where each facet is an $(n-1)$ flat. Let $\mathcal{A}$ be an arrangement of $k-1$ hyperplanes in $\mathbb{R}^{n-1}$ that generates a maximal number of regions. Each one of the regions generated by $\mathcal{A}$ is certainly a polytope contained in $\mathbb{R}^{n-1}$, so since there are finitely many polytopes each with finitely many vertices, let $\epsilon$ be the minimal distance between any two vertices within the same polytope. Let the $i^{th}$ hyperplane in $\mathcal{A}$ be defined as $\{x \in \mathbb{R}^{n-1} \mid a_i^T x = b_i\}$. Then we can define $\mathcal{P}_{i+1} := \{x \in \mathbb{R}^n \mid b_i - \epsilon/3 \leq (0, a_i)^T x \leq b_i + \epsilon/3\}$. Thus the $(n-1)$-flat that describes each

65  facet of $\mathcal{P}_1$ remains broken up into $R(n-1, k-1) = \Omega(k^{n-1})$ disjoint convex components. Each
66  of these exists on the boundary of the union of $\mathscr{P}$. □

67  Now we can restate and prove our theorems regarding the efficient boundary decompositions of
68  polyhedral complexes.

69  **Theorem 3.1.** *Given a polyhedral complex, $\mathscr{P} = \{\mathcal{P}_1, \ldots \mathcal{P}_k\}$, where $\mathcal{P}_i$ is defined as the intersec-*
70  *tion of $m_i$ closed halfspaces. Let $M = \sum_i m_i$, and let $x_0$ be a point contained by at least one such*
71  *$\mathcal{P}_i$. Then the boundary of $\bigcup_{i \in [k]} \mathcal{P}_i$ is represented by at most $M$ $(n-1)$-dimensional polytopes.*
72  *There exists an algorithm that can compute this boundary in $\mathcal{O}(poly(n, M, k))$ time.*

73  *Proof.* Let $Z = \bigcup_{i \in [k]} \mathcal{P}_i$. Let $F_{i,j}$ refer to the $j^{th}$ facet of $\mathcal{P}_i$, and let $\mathcal{F}_i$ be the set of facets of $\mathcal{P}_i$
74  that are not facets of any other $\mathcal{P}_j$. Then, letting $T = \bigcup_{i \in [k]} \mathcal{F}_i$. We claim that the boundary of $Z$ is
75  exactly $T$.

76  Without loss of generality, assume that $Z$ is a single connected component, in the topological sense. If
77  $Z$ were multiple connected components, then we could handle each of them in turn. To demonstrate
78  that $T$ is the boundary of $Z$ we need to show that for any $x \in T$ that points (i), (ii) of definition 1
79  hold, and that condition (ii) fails for any point $y \in Z \setminus T$.

80  To demonstrate point (i) above, we note that $x \in \mathcal{P}_i$ for at least one $\mathcal{P}_i$. By assumption each $\mathcal{P}_i$ has
81  a nonempty interior, and thus contains some point $y \in \mathcal{P}_i$ for which a neighborhood $N(y) \subset \mathcal{P}_i$.
82  Thus if $\mathcal{P}_i$ is given as an $H$-polytope of the form $\{x \mid Ax \leq b\}$, then $Ay < b$. Since $\mathcal{P}_i$ is
83  convex, then any convex combination between $x, y$ is contained in $\mathcal{P}_i$, and in fact for all $\lambda \in [0, 1)$,
84  $A(\lambda x + (1 - \lambda)y) < b$. Certainly any point $z$ such that $Az < b$ has a neighborhood $N(z)$ contained
85  in $\mathcal{P}_1$.

86  Proving that $x \in T$ satisfies point (ii) is more complicated. Let $\mathcal{Q}$ be a facet containing $x$, and let
87  $\mathcal{P}_i$ be a polytope containing $\mathcal{Q}$. Let $H$ be the hyperplane containing $\mathcal{Q}$. Then for all $j \neq i$, $\mathcal{P}_i \cap \mathcal{P}_j$
88  is either the empty set or resides in a face of $\mathcal{P}_j$ of dimension at most $(n-2)$. A standard result
89  about polytopes states that if $\mathcal{Q}$ is an $(n-1)$ dimensional polytope, it can be defined by the set
90  $\{x \mid A^= x = b^= \wedge A^- x \leq b^-\}$ where $A^=$ has rank 1. Additionally there exists a point $y \in \mathcal{Q}$ such
91  that $A^- y < b$ [5]. Then every point along the open line segment $(x, y)$ is contained in the relative
92  interior of $\mathcal{Q}$, and by definition cannot be contained in any face of $\mathcal{P}_j$ for $j \neq i$. Further, since the
93  relative interior of $\mathcal{Q}$ is open, every point $w$ along $(x, y)$ is contained in a neighborhood $N(w)$, with
94  restriction to $H$ $N(w)_{|H}$. Then certainly $N(w)_{|H} \subseteq relInt(\mathcal{Q}) \subset \mathcal{Q}$, which implies that $N(w)_{|H}$
95  is disjoint from $\cup_{j \neq i} \mathcal{P}_j$.

96  Let $H^-$ be the closed halfspace defined by $H$ containing $\mathcal{P}_i$, then $N(w) \cap (H^-)^c$ is both open and
97  disjoint from $\mathcal{P}_i$ in addition to being disjoint from $\mathcal{P}_j$ for all $j \neq i$. Let $c$ be a point in $N(w) \cap Z^c$,
98  such that the open line segment between $(w, c)$ is contained in $N(w) \cap Z^c$. We now restrict our
99  attention to the 2-dimensional linear subspace of $\mathbb{R}^n$ containing $x, w, c$, denoted as $V$. Each $\mathcal{P}_{j|V}$ is
100  either the emptyset or a polytope containing $x$. Let $\mathcal{U}_{j|V}$ be the set of these 2-d restricted polytopes
101  containing $x$, and note that each $\mathcal{U}_{j|V}$ intersects with $\mathcal{P}_{i|V}$ only at $x$. Because each element of $\mathcal{U}_{|V}$
102  intersects with $\mathcal{P}_{i|V}$ only at $x$, there must a hyperplane $H_j$, (line in $V$) passing through $x$ separating
103  each element of $\mathcal{U}_{|V}$ and $c$. Let $H_j^+$ be the closed halfspace defined by $H_j$ containing $c$. Then $\cap H_j$
104  defines a polytope $\mathcal{S}$ that only intersects with $\mathcal{P}_{\rangle}$ at $x$. The line segment between $(x, c)$ lies inside $\mathcal{S}$
105  and thus does not intersect any $\mathcal{P}_{j|V}$ for $j \neq i$. $(x, c)$ also lies strictly on one side of the hyperplane
106  $H$ that $\mathcal{Q}$ resides in, and thus every point along $(x, c)$ is not contained in $\mathcal{P}_i$. Hence, $(x, c)$ is not
107  contained in $Z$, as desired.

108  Finally, to show that there is no point $y$ in the boundary of $Z$ that not contained in $T$. It suffices
109  to show that $Z \setminus T$ is open, as if this were the case, then any $y \in Z \setminus T$ would be contained in a
110  neighborhood $N(y) \subseteq Z \setminus T$ and thus fail to meet condition (ii) of the definition of the boundary.
111  Let $x \in Z \setminus T$. Then $x$ is contained in the interior of some $\mathcal{P}_i$ or it is contained in a facet contained in
112  both $\mathcal{P}_i, \mathcal{P}_j$, for some $i, j$. This follows from the fact that $x$ either is contained in a facet of some $\mathcal{P}_i$
113  or not. If not, $x$ is strictly in the interior of some $\mathcal{P}_i$ and is contained in a neighborhood $N(x) \subset \mathcal{P}_i$.
114  If so, then $x$ needs to be contained in a facet, $F_{i,j}$ of $\mathcal{P}_i$ and $\mathcal{P}_j$, else $x \in T$. Either $x$ is contained in
115  the relative interior of $F_{i,j}$ or not. If so, then a neighborhood of $x$, $N(x)$, is bisected by $F_{i,j}$, where
116  each half is contained in either $\mathcal{P}_i$ or $\mathcal{P}_j$. If not, then $x$ needs to be contained in a facet of some $\mathcal{P}_m$,
117  for $m \neq i, j$, because it needs to be contained in some other facet of $\mathcal{P}_i$. This other facet needs to be

118  a facet of some $\mathcal{P}_m$ because otherwise it would be contained in $T$ and certainly $\mathcal{P}_i \cap \mathcal{P}_j = F_{i,j}$ such
119  that $m \neq j$. We repeat this process until we have enumerated all facets containing $x$, of which there
120  are at most $\binom{k}{2}$. There are then at most $k$ polytopes containing $N(x)$, and their union contains $N(x)$.
121  Thus $Z \setminus T$ is open.

122  To demonstrate that $T$ is represented by at most $M$ polytopes and that $T$ can be computed in
123  $\mathcal{O}(poly(n, M, k))$ time, note that each polytope $\mathcal{P}_i$ has at most $m_i$ facets, and not all of these are
124  included in $T$. Thus the number of facets, and hence polytopes, that define $T$ is at most $\sum m_i = M$.
125  Enumerating each of these polytopes can be done in time linear in $M$. To compare if two facets are
126  equivalent, one can find a point $y \in F_{i,j}$ such that it is in the relative interior of $F_{i,j}$. Such a point can
127  be found in polynomial time using a linear program. Since $\mathscr{P}$ is a polyhedral complex, if such a $y$ is
128  contained in $F_{i,j}$ and $F_{i',j'}$ then $F_{i,j} = F_{i',j'}$. There are at most $\binom{M}{2}$ facets, so $T$ can be determined
129  in time polynomial in $n, M, k$.

130                                                                                                                $\square$

# C  Proofs of Correctness for GeoCert

132  In this section we expand upon the graph theoretic interpretation of GeoCert and prove its correctness.
133  Recall the setup: given a polyhedral complex $\mathscr{P}$, which can be viewed as a bipartite graph of $n$-
134  dimensional polytopes and their $(n-1)$-dimensional faces, some of which are labeled as 'boundary'
135  facets, our goal is to return the boundary facet which admits minimal distance to a fixed point $x_0$.
136  In our primary discussion we replaced 'distance' with a 'potential' function. Formally, we let our
137  pointwise potential to be some function $\phi : \mathbb{R}^n \to \mathbb{R}$, and the facetwise potential, $\Phi : \mathcal{P}(\mathbb{R}^n) \to$
138  $(\mathbb{R} \cup \{+\infty\})$ to be defined as

$$\Phi(\mathcal{F}) = \begin{cases} +\infty, & \text{if } \mathcal{F} = \emptyset \\ \min_{y \in \mathcal{F}} \phi(y), & \text{otherwise} \end{cases} \tag{9}$$

139  Certainly, letting $\phi(y) := ||y - x_0||$ and finding the boundary facet with minimal potential $\Phi$ is
140  equivalent to finding the facet with minimal distance to $x_0$. However, this choice of $\phi$ is not the
141  only valid one for which GeoCert will provide the corect answer to the centered Chebyshev ball
142  problem. To this end, we provide a sufficient condition on a pointwise potential function $\phi$ such that
143  GeoCert will still provide the correct answer. We can then demonstrate that any potential function
144  satisfying this property will cause GeoCert to return the correct answer. Finally we can show that the
145  $\ell_p$-distance potential satisfies these properties, and that the lipschitz potential described in Section 5
146  also satisfies this property.

147  **Definition 1.** *Given a potential function $\phi$ defined only on the set of points contained in a polyhedral*
148  *complex $\mathscr{P}$, we let $\eta_v(t) := \phi(x_0 + t \cdot v)$ for any vector $v$ and any positive scalar $t > 0$. Then we*
149  *say that $\phi$ is **ray monotonic** if for every $v, t > 0$, $\dfrac{\delta \eta}{\delta t}(t) \geq 0$.*

150  With this definition in hand, we can prove a structural invariant of the operation of GeoCert that will
151  directly prove the claim of correctness.

152  **Lemma C.1.** *For any polyhedral complex $\mathscr{P}$ point $x_0$, and ray-monotonic potential $\phi$, let $\mathcal{F}_i$ be the*
153  *facet popped at the $i^{th}$ iteration of GeoCert. Then for all $i < j$, $\Phi(\mathcal{F}_i) \leq \Phi(\mathcal{F}_j)$.*

154  *Proof.* We proceed by induction. In the base case we only consider the first and second iteration.
155  Supposing without loss of generality that $x_0$ is contained in exactly one polytope $\mathcal{P} \in \mathscr{P}$. Then the
156  initial set of facets added to the priority queue is exactly the set of facets of $\mathcal{P}$, which we denote as
157  $\{\mathcal{F}_\mathcal{P}(1), \mathcal{F}_\mathcal{P}(2), \ldots, \mathcal{F}_\mathcal{P}(k)\}$ which are ordered by potential, without loss of generality.

158  At the first iteration, $\mathcal{F}_\mathcal{P}(1)$ is popped, and a new polytope $\mathcal{S}$ is opened. The set of facets of added
159  to the priority queue $Q$, also ordered by potential, is $\{\mathcal{F}_\mathcal{S}(1), \mathcal{F}_\mathcal{S}(2), \ldots, \mathcal{F}_\mathcal{S}(k)\}$. We would like
160  to show that whichever facet $\mathcal{F}_2$, is popped at iteration 2 must have that $\Phi(\mathcal{F}_2) \geq \Phi(\mathcal{F}_\mathcal{P}(1))$. As,
161  by definition, for all $i > 1$, $\Phi(\mathcal{F}_\mathcal{P}(1)) \leq \Phi(\mathcal{F}_\mathcal{P}(i))$ it suffices to show that any facet $\mathcal{F}_\mathcal{S}$ of $\mathcal{S}$
162  added to the priority queue must have $\Phi(\mathcal{F}_\mathcal{P}(1)) \leq \Phi(\mathcal{F}_\mathcal{S})$. For any facet of $\mathcal{F}_\mathcal{S}$, we have that
163  $\Phi(\mathcal{F}_\mathcal{S}) := \min_{y \in \mathcal{F}_\mathcal{S}(1))} \phi(y)$. Letting $y_{min}$ be an element of the argmin of this minimum, we utilize the

164  ray-monotonic property of $\phi$. We let $v = y_{min} - x_0$ and note that $\Phi(\mathcal{F}_{\mathcal{S}}) = \phi(x_0 + v)$. As $y_{min}$ is
165  not contained in the interior of $\mathcal{P}$, there must exist some $t \in [0, 1]$ such that $x_0 + tv$ lies in a facet of
166  $\mathcal{P}$. By definition $\Phi(\mathcal{F}_{\mathcal{P}}(1)) \leq \phi(x_0 + tv) \leq \phi(x_0 + v)$, where the first inequality comes from the
167  definition of $\Phi$, and the second inequality comes from the ray-monotonicity of $\phi$. This concludes the
168  base case.

The inductive step follows by a similar argument. Suppose the claim holds up to iteration $i - 1$. At
the $i^{th}$ iteration we pop facet $\mathcal{F}_i$, open up a previously-unseen polytope $\mathcal{S}$, and add a set of facets
each corresponding to another unseen polytope: hence no potential facet added has been previously
added to the priority queue. Again, considering any new facet $\mathcal{F}_{\mathcal{S}}$ and the argmin of its potential

$$y_{min} \in \arg\min_{y \in \mathcal{F}_{\mathcal{S}}} \phi(y)$$

169  we note that $y_{min}$ is not contained in the interior of any of the set of seen polytopes $C$. Then again
170  letting $y_{min} = x_0 + v$, there exists some $t \in (0, 1]$ such that $x_0 + tv$ lies in some facet $\mathcal{G}$ that is
171  contained in the priority queue at iteration $(i - 1)$. Since $\Phi(\mathcal{F}_{(i-1)}) \leq \Phi(\mathcal{G}) \leq \phi(y_{min}) = \Phi(\mathcal{F}_{\mathcal{S}})$,
172  we maintain our structural invariant and the proof is complete.  □

173  **Theorem C.1.** *For a fixed polyhedral complex $\mathscr{P}$, a fixed input point $x_0$ and a potential function $\phi$*
174  *that is ray-monotonic, GeoCert returns a boundary facet with minimal potential $\Phi$.*

175  *Proof.* Leveraging Lemma C.1, we note that since we only pop facets in non-decreasing order, the
176  first 'boundary facet' that is popped will be a boundary facet with minimal potential.  □

177  Now we simply need to show that both choices of potential function discussed satisfy the ray-
178  monotonicity property.

179  **Corollary C.1.** *The distance potential, $\phi_{lp}(y) := ||y - x_0||$ satisfies ray-monotonicity and Geocert*
180  *using this as a potential returns the minimal distance boundary facet.*

181  *Proof.* We fix a vector $v$ and any scalar $t > 0$. We define

$$\eta_v(t) := ||(x_0 + tv) - x_0|| = |t| \cdot ||v|| = t \cdot ||v|| \tag{10}$$

182  Then $\dfrac{\delta \eta_v}{\delta t} = ||v|| \geq 0$ for all $t > 0, v$.  □

183  **Corollary C.2.** *For a PLNN $f : \mathbb{R}^n \to \mathbb{R}^k$ and a point $x_0$, let $i := \arg\max_j f_j(x_0)$. Let $DR(x_0) =$*
184  *$\{x \mid \arg\max_j f(z) = i\}$. Define $g_j(x) = f_i(x) - f_j(x)$ for all $j \neq i$, and let $L_j$ be a bound on the*
185  *$\ell_q$ lipschitz constant of $g_j$:*

$$|g_j(x) - g_j(y)| \leq L_j ||x - y||_p \qquad \forall x, y \in DR(x_0) \tag{11}$$

186  *then the potential*

$$\phi_{lip,j}(y) := ||y - x_0||_p + \frac{g_j(y)}{L_j} \tag{12}$$

$$\phi_{lip}(y) := \min_j \phi_{lip,j}(y) \tag{13}$$

187  *satisfies ray-monotonicity and Geocert using this as a potential returns the minimal distance boundary*
188  *facet.*

189  *Proof.* We prove the ray-monotonicity for each $\phi_j$ and then demonstrate that this holds for their
190  minimum as well. First we note that for every point $x \in DR(x_0)$ has that $g_j(x) \geq 0$. Fixing some
191  $\phi_j, v$, and $t > 0$ such that $x_0 + tv \in DR(x_0)$, we consider

$$\eta_{j,v}(t) := \phi_{lip,j}(x_0 + tv) = t||v||_p + \frac{g_j(x_0 + tv) - g_j(x_0)}{L_j} \tag{14}$$

which has derivative

$$\frac{\delta \eta_{j,v}}{\delta t}(x_0 + tv) = ||v||_p + \frac{1}{L_j}\frac{\delta g_j}{\delta t}(x_0 + tv) \tag{15}$$

$$= ||v||_p + \frac{1}{L_j}\langle v, \nabla g_j(x_0 + tv)\rangle \tag{16}$$

$$\geq ||v||_p - \frac{1}{L_j}||V||_p|||\nabla g_j(x_0 + tv)||_q \tag{17}$$

$$\geq ||v||_p(1 - 1) \tag{18}$$

$$\geq 0 \tag{19}$$

Where the first inequality comes from Hölder's inequality, and the second inequality comes from the fact that the norm of the gradient is bounded by the lipschitz constant. And since the minimum of monotonically increasing functions is also monotonically increasing, $\phi$ is ray-monotonic. This implies that GeoCert returns the minimal potential facet. However, note that along any boundary facet $\mathcal{F}_{bound}$, there exists a $j$ such that $g_j(y) = 0 \forall y \in \mathcal{F}_{bound}$. Since each $g_j(y) \geq 0$ for all $y \in DR(x_0)$ for any $y \in \mathcal{F}_{bound}$, $\phi(y) = ||x_0 - y||_p$. In other words, this potential function is equivalent to the $\ell_p$ potential along the decision boundary. Hence the first 'boundary facet' popped is the boundary facet with minimal $\ell_p$ distance, as desired. $\qquad\square$

**Remarks:** Recall that as a subroutine, GeoCert using $\phi_{lip}$ as a potential, must compute $\Phi_{lip}(\mathcal{F})$ for each possible facet $\mathcal{F}$ to be added to the priority queue. This amounts to solving the following optimization problem

$$\Phi_{lip}(\mathcal{F}) := \min_{y \in \mathcal{F}}\left(||y - x_0||_p + \min_{j \neq i}\frac{g_j(y)}{L_j}\right) \tag{20}$$

Along each piecewise linear region of a PLNN, certainly $f$ is a linear function, as is $g_j$. Hence, computing the minimum of $\phi_{lip,j}$ across a facet requires as much computation time as computing the $\ell_p$ projection to a facet. Since $\min_{j \neq i} g_j(y)$ is a pointwise minimum and hence not convex, computing $\Phi_{lip}$ is no longer computable by a single convex program. However one can minimize this for each $\phi_{lip,j}$ and return the overall minimum. This now requires multiple convex programs per facet. We find that (i) using a warm-start for our optimizations allows the second-through-final to finish much more quickly than the initial optimization, and (ii) a variant of GeoCert can be used where the facet-wise potential is replaced with a polytope-wise potential. Under this formulation, the number of optimizations per polytope with $m$ constraints goes from $m$, in the case of the $\ell_p$ potential, to $m + (k - 1)$ where $k$ is the number of logits: we simply need to compute the feasibility of each facet ($m$ linear programs), to determine the neighbors of the right vertices in the graph, and $(k - 1)$ optimizations to compute the polytope-wise potential.

Finally, we remark about the efficient computation of $L_j$. Under a fixed domain $\mathcal{D}$, if a lower and upper bound to each input to each ReLU of the neural net is known, a nontrivial upper bound to each $L_j$ can be computed with as much computation as is required by eight forward passes through the PLNN [8]. Indeed, by leveraging $\phi_{lip}$ as a potential, one can effectively propagate the lower-bound to pointwise robustness as computed by Fast-Lip: instead of computing a certifiable lower bound only on $f$ evaluated at $x_0$, as Fast-Lip does, the certifiable lower bound is now computed across every facet in the 'frontier set' which expands outwards as GeoCert runs. This allows for Fast-Lip to converted into continually increasing lower bound.

## D    Polyhedral Complex Properties

Here we will restate and prove the lemmas regarding iterative construction of polyhedral complices, and other useful tools when considering the centered Chebyshev ball contained in a polyhedral complex.

**Lemma 3.3.** *Given an arbitrary polytope $\mathcal{P} := \{x \mid Ax \leq b\}$ and a hyperplane $\mathcal{H} := \{x \mid c^T x = d\}$ that intersects the interior of $\mathcal{P}$, the two polytopes formed by the intersection of $\mathcal{P}$ and the each of closed halfpsaces defined by $\mathcal{H}$ are PC.*

Figure 3: Pictorial aid for Lemma 3.4.

231  *Proof.* Let $\mathcal{H}^+ := \{x \mid c^T x \geq d\}$ and $\mathcal{H}^- := \{x \mid c^T x \leq d\}$, with $\mathcal{P}^+ := \mathcal{P} \cap \mathcal{H}^+$ and
232  $\mathcal{P}^- := \mathcal{P} \cap \mathcal{H}^-$. Then each of $\mathcal{P}^+, \mathcal{P}^-$ are polytopes with nonempty interior and their intersection
233  is exactly $\mathcal{P} \cap \mathcal{H}$, which is a face of both $\mathcal{P}^+, \mathcal{P}^-$.  □

234  **Lemma 3.4.** *Let $\mathcal{P}, \mathcal{Q}$ be two PC polytopes and let $H_\mathcal{P}, H_\mathcal{Q}$ be two hyperplanes that define two*
235  *closed halfspaces each, $H_\mathcal{P}^+, H_\mathcal{P}^-, H_\mathcal{Q}^+, H_\mathcal{Q}^-$. If $\mathcal{P} \cap \mathcal{Q} \cap H_\mathcal{P} = \mathcal{P} \cap \mathcal{Q} \cap H_\mathcal{Q}$ then the subset of the*
236  *four resulting polytopes $\{\mathcal{P} \cap H_\mathcal{P}^+, \mathcal{P} \cap H_\mathcal{P}^-, \mathcal{Q} \cap H_\mathcal{Q}^+, \mathcal{Q} \cap H_\mathcal{Q}^+\}$ with nonempty interior forms a*
237  *polyhedral complex.*

238  *Proof.* Let $F = \mathcal{P} \cap \mathcal{Q}$, which by definition is a face of both $\mathcal{P}, \mathcal{Q}$. Without loss of generality we
239  can align the hyperplanes $H_\mathcal{P}, H_\mathcal{Q}$ such that $F \cap H_\mathcal{Q}^+ = F \cap H_\mathcal{P}^+$. For ease of notation, we'll let
240  $\mathcal{P}^+$ denote $\mathcal{P} \cap H_\mathcal{P}^+$, and similarly for $\mathcal{P}^-, \mathcal{Q}^+, \mathcal{Q}^-$. If $H_\mathcal{P}$ does not intersect the interior of $\mathcal{P}$, then
241  exactly one of $\mathcal{P}^+, \mathcal{P}^-$ has empty interior and can be ignored. Otherwise, by lemma 3.3, $\mathcal{P}^+, \mathcal{P}^-$ are
242  PC, and likewise for $\mathcal{Q}^+, \mathcal{Q}^-$. To handle the cross-terms we proceed by cases. Letting $S = F \cap H_\mathcal{P}$,
243  we handle the following four cases: (i) $S = \emptyset$, (ii) $S$ is a face of $F$, (iii) $S = F$, or (iv) none of the
244  above.

245  (i): In the case that $S = \emptyset$, then either $\mathcal{P}^+ \cap F$ or $\mathcal{P}^- \cap F$ is empty. Likewise for $\mathcal{Q}^+ \cap F, \mathcal{Q}^- \cap F$.
246  Assume without loss of generality that $\mathcal{P}^+ \cap F = \mathcal{Q}^+ \cap F = \emptyset$. Then certainly $\mathcal{P}^+$ is disjoint from
247  $\mathcal{Q}$ and therefore both $\mathcal{Q}^+, \mathcal{Q}^-$. Likewise for the interaction between $\mathcal{Q}^+$ and $\mathcal{P}^-, \mathcal{P}^+$. Finally, since
248  $S = \emptyset$, $F$ is a face of both $\mathcal{P}^-$ and $\mathcal{Q}^-$ and $\mathcal{P}^- \cap \mathcal{Q}^- = F$, hence they are PC.

249  (ii): In the case that $S$ is a face of $F$, we label this face $G$. First note that $F$ needs to be fully contained
250  by either $F \cap H_\mathcal{P}^+$ or $F \cap H_\mathcal{P}^-$. Thus $F$ is either a face of $\mathcal{P}^+$ or $\mathcal{P}^-$, where we can assume without
251  loss of generality that it is a face of $\mathcal{P}^-$. Similarly, assume $F$ is a face of $\mathcal{Q}^-$, implying that $\mathcal{P}^-$ and
252  $\mathcal{Q}^-$ are PC. By this assumption, $\mathcal{P}^+ \cap F = G$. Note that $G$ is a face of $\mathcal{P}^+$. Since $G$ is a face of $F$, it
253  is also a face of $\mathcal{Q}^-$, and $\mathcal{P}^+ \cap \mathcal{Q}^- = G$, which is a face of each of them and therefore $\mathcal{P}^+$ and $\mathcal{Q}^-$
254  are PC. Likewise for $\mathcal{Q}^+$ and $\mathcal{P}^-$. Finally note that since $\mathcal{P}^+ \cap F = \mathcal{Q}^+ \cap F = G$, implying that
255  $\mathcal{P}^+ \cap \mathcal{Q}^+ = G$, hence $\mathcal{P}^+$ and $\mathcal{Q}^+$ are PC.

256  (iii): If $S = F$, then we can assume without loss of generality that $\mathcal{P}^- = \mathcal{P}$ and $\mathcal{P}^+ = F$, and
257  similarly for $\mathcal{Q}$. Then since $\mathcal{Q}^+ = \mathcal{P}^+ = F$ they do not have nonempty interior and can be ignored.
258  By definition $\mathcal{P}^-$ and $\mathcal{Q}^-$ are PC, and $\mathcal{P}^-, \mathcal{Q}^+$ are as well. (iv): In the final case, $S$ is neither the
259  emptyset, $F$, nor a face of $F$. Then $F \cap H_\mathcal{Q}^+$ and $F \cap H_\mathcal{Q}^-$ are both nonempty polytopes with the
260  same dimensionality as $F$. Letting $S^+ = F \cap H_\mathcal{Q}^+$, and defining $S^-$ likewise, note that $S$ is a face of
261  $S^+, S^-$, by the same argument used in 3.3. Since $F$ is a face of $\mathcal{P}$, $S^+$ is a face of $\mathcal{P}^+$ and likewise
262  for $\mathcal{Q}^+$. And since $\mathcal{P}^+ \subseteq \mathcal{P}$, $\mathcal{P}^+ \cap \mathcal{Q}^+ \subseteq \mathcal{P} \cap \mathcal{Q} = F$. But $\mathcal{P}^+ \cap F = S^+$ and $\mathcal{Q}^+ \cap F = S^+$, thus
263  $\mathcal{P}^+ \cap \mathcal{Q}^+ = S^+$. Hence $\mathcal{P}^+$ and $\mathcal{Q}^+$ are PC. Likewise for $\mathcal{P}^-$ and $\mathcal{Q}^-$. Since $\mathcal{P}^+ \cap \mathcal{Q}^- = S$ and $S$
264  is a face of $S^+, S^-$, it is a face of both $\mathcal{P}^+, \mathcal{Q}^-$ and the two are PC. Likewise for $\mathcal{P}^-$ and $\mathcal{Q}^+$.

265  □

266  **Lemma 3.5.** *Let $\mathscr{P} = \{\mathcal{P}_1, \ldots \mathcal{P}_k\}$ be a polyhedral complex and let $\mathcal{D}$ be any polytope. Then the*
267  *set $\{\mathcal{P}_i \cap \mathcal{D} \mid \mathcal{P}_i \in \mathscr{P}\}$ also forms a polyhedral complex.*

268 *Proof.* Letting $H_j$ be the hyperplanes that compose $\mathcal{D}$, i.e., $\mathcal{D} = \bigcap_j H_j$. Then it suffices to show
269 that $\{\mathcal{P}_i \cap H_j \mid \mathcal{P}_i \in \mathscr{P}\}$ is a polyhedral complex, as we can repeat this iteratively for each $H_j$. This
270 is equivalent to stating that for each $\mathcal{P}_i, \mathcal{P}_j \in \mathscr{P}$ with nonempty intersection, $\mathcal{P}_i \cap H_j$ and $\mathcal{P}_j \cap H_j$
271 are PC. This follows from a direct application of Lemma 3.4. $\qquad\square$

272 **Lemma D.1.** *Let $\mathcal{P}$, $\mathcal{Q}$ be polytopes whose intersection is $(n-d)$ dimensional, for some $d \geq 2$, and*
273 *let $x_0 \in \mathcal{P}$, with $B_t(x_0)$ the largest $\ell_p$-norm ball centered at $x_0$ contained in $\mathcal{P} \cup \mathcal{Q}$. Then $B_t(x_0)$ is*
274 *contained entirely in $\mathcal{P}$.*

275 *Proof.* First we state an equivalent representation of $B_t(x_0)$,

$$B_t(x_0) = \bigcup_{\{z \mid ||x_0 - z|| \leq t\}} B_d(z) \quad \text{for } d = (t - ||x_0 - z||) \tag{21}$$

Certainly the $\subseteq$ inclusion holds by setting $z = x_0$ and the $\supseteq$ inclusion holds by the triangle inequality.
Now let's assume that $\mathcal{P} \cap \mathcal{Q}$ is nonempty and contained in an $(n-2)$-dimensional linear subspace,
$H$. Suppose for the sake of contradiction that $r > 0$ for

$$r := \sup_{x \in \mathcal{P} \cap \mathcal{Q}} t - ||x - x_0||$$

276 and $z$ is defined as some point in $\mathcal{P} \cap \mathcal{Q}$ that attains this supremal distance. Such a $z$ must exist
277 because $\mathcal{P} \cap \mathcal{Q}$ is closed. Then $B_r(z) \subseteq B_t(x_0) \subseteq (\mathcal{P} \cup \mathcal{Q}) \subseteq H$. But $B_r(z)$ is contains some $\ell_2$
278 ball, regardless of our choice of norm, contradicting the previous chain of inclusions. Thus $r \leq 0$,
279 indicating that $B_t(x_0) \subseteq \mathcal{P}$. $\qquad\square$

# E   Geometry of Piecewise Linear Neural Networks

281 In this appendix we restate and prove our theorems regarding the geometry of PLNN's. Specifically,
282 we prove our lemma which describes that each ReLU configuration defines a polytope and, in general
283 position, its facets correspond to exactly one ReLU being flipped. Then we prove that the decision
284 region forms a polyhedral complex.

## E.1   Computing the linear region of neural networks

286 First we prove this lemma:

287 **Lemma 4.1.** *For a given neuron configuration $A$, the following are true about $\mathcal{P}_A$,*

288     *(i) Unless $\mathcal{P}_A = \mathbb{R}^n$ or $\emptyset$, there exists a neuron configuration $B$ such that $\mathcal{P}_A \cap \mathcal{P}_B \neq \emptyset$.*

289     *(ii) $\mathcal{P}_A$ is a polytope, and for all layers $i$, $f^{(i)}(x)$ is linear in $x$ for all $x \in \mathcal{P}_A$.*

290 *Proof.* **Item (i)**: This is trivial as certainly every point in the domain corresponds to at least one
291 neuron configuration. If both $\mathcal{P}_A$ and $\mathcal{P}_A^c$ are not the empty set, then their intersection is nonempty.
292 But $\mathcal{P}_A^c$ is composed of a union of at least one piecewise linear region, at least one of which must
293 intersect $\mathcal{P}_A$.

**Item (ii)**: This is easy to see by simply writing down the polytope $\mathcal{P}_A$ and its corresponding linear
function. For neuron configuration $A$, we partition $A$ into $A_1, A_2, \ldots A_{l-1}$, with $A_i$ corresponding
to the neuron configuration at the $i^{th}$ layer. Then letting $\Lambda_i$ be a fixed matrix to replace each ReLU in
the network, defined as $\Lambda_i := \text{diag}(A_i)$ we note that

$$f^{(i)}(x) = \begin{cases} W_i x + b_i, & \text{if i = 1} \\ W_i \sigma(\Lambda_i)(f^{(i-1)}(x)) + b_i, & \text{if } i > 1 \end{cases}$$

294 Hence, as $\sigma(\Lambda_i)$ is constant across all points with neuron configuration $\mathcal{A}$, $f$ is a composition of linear
295 functions and must be linear everywhere with that neuron configuration. To define the polytope $\mathcal{P}_A$,
296 we note that each neuron adds one linear constraint to the polytope. Let us write down each of these
297 constraints exactly. Since each $f^{(i)}(x)$ is linear, it can be written as $V_i x + c_i$ for some $V_i, c_i$. Recalling
298 that $f^{(i)}(x)$ is the input to the $i^{th}$ ReLU layer, the constraints are of the form $f^{(i)}(x) \underset{\sim}{?} 0$ where
299 $\underset{\sim}{?}$ is the comparator $\geq, \leq, =$ for $A_{i,j}$ being $1, -1, 0$ respectively. This can be encoded efficiently

300 by multiplying the lefthand side by $-\Lambda_i$, so the total constraint becomes $\Lambda_i(V_i x + c_i) \geq 0$. We
301 remark that $\Lambda_i$ can be computed with a single forward pass of the network, and each $V_i$ and $c_i$ can be
302 computed with a two matrix multiplications, one of which is a diagonal matrix.

303 □

## E.2 PLNN's Form Polyhedral Complices

305 We can now prove our main theorem regarding the linear regions of a PLNN.

306 **Theorem 4.2.** *The collection of $\mathcal{P}_A$ for all $A$, such that $\mathcal{P}_A$ has nonempty interior forms a polyhedral*
307 *complex. Further, the decision region of $F$ at $x_0$ also forms a polyhedral complex.*

308 *Proof.* Let $\mathscr{P}_{i,j}$ denote the set of polytopes generated by neuron configurations of all neurons in
309 layer $k < i$, and the first $j$ neurons in layer $i$. Let $\mathscr{P}_{i,0}$ refer to the set of polytopes generated by
310 neuron configurations from all neurons in layer $k < i$. We'll prove the theorem by induction across $i$,
311 with an inner induction on $j$.

312 As a base case, consider only the first layer $f^{(1)}(x)$. Examining only neuron $j$ of the first layer,
313 note that $f^{(1)}(x)_j = W_{1,j} x + b_{1,j}$ implying that the, unless $W_{1,j} = 0$, the set of inputs $x$ for which
314 $f^{(1)}(x)_j = 0$ is exactly a hyperplane, which we shall denote $H_j$. Then we can perform a second,
315 interior, induction across the neurons of the first layer of $f$.

316 The first neuron in the first layer separates $\mathbb{R}^n$ into two closed halfspaces, such that $\mathscr{P}_{1,1}$ is PC. Now
317 assume that $\mathscr{P}_{1,k}$ is PC. Consider now the addition of the $(k+1)^{th}$ neuron to generate $\mathscr{P}_{1,k+1}$.
318 In particular, if $\mathscr{P}_{1,k}$ is generated by considering the arrangement of hyperplanes $H_1, \ldots H_k$, then
319 $\mathscr{P}_{1,k+1}$ is $\mathscr{P}_{1,k}$ with the addition of hyperplane $H_{k+1}$. Letting $\mathcal{PQ}$ be two PC polytopes in $\mathscr{P}_{1,k}$,
320 we can let $H_{k+1}$ define $H_\mathcal{P}$ and $H_\mathcal{Q}$ and apply lemma 3.4 to demonstrate that the polytopes generated
321 by this intersection remain PC. This concludes the base case of the outer induction.

322 Now let's assume that for any layer $k$, $\mathscr{P}_{k,0}$ is a polyhedral complex. Consider the difference between
323 $\mathscr{P}_{k,0}$ and $\mathscr{P}_{k,1}$. Let $G_1$ refer to the set of points $x$ for which $f_1^{(k)}(x) = 0$, i.e. the first neuron of
324 layer $k$ has pre-ReLU value exactly zero. Now by 4.1 part **??**, $f^{(k)}(x)_1$ is linear in each $\mathcal{P}_A \in \mathscr{P}_{k,0}$.
325 Thus for each such $\mathcal{P}_A$, $G_1 \cap \mathcal{P}_A$ is either the emptyset or a hyperplane, $H_A$. Any two polytopes
326 $\mathcal{P}_A, \mathcal{P}_B$ contained in $\mathscr{P}_{k,0}$ with nonempty intersection, by inductive assumption, must be PC. If
327 $H_A \cap F \neq \emptyset$, then certainly $G_1 \cap \mathcal{P}_B \neq \emptyset$ and thus there must be some hyperplane $H_B$ such that
328 $H_B = \mathcal{P}_B \cap G_1$. Since $F \cap G_1 = H_A \cap F$ and $F \cap G_1 = H_B \cap F$, we meet the criteria to apply
329 lemma 3.4 and thus the polytopes generated by the addition of $G_1$ remain PC.

330 To conclude the proof of the first statement in the theorem, assume that $\mathscr{P}_{k,j}$ is PC. Then consider
331 the addition of the $(j+1)^{th}$ neuron of layer $k$. Let $G_{j+1}$ refer to the set of points for which
332 $f_{j+1}^{(k)}(x) = 0$. Note that $f_{j+1}^{(k)}$ is linear across each $\mathcal{P}_A \in \mathscr{P}_{k,0}$, since we just as well could have
333 initially incorporated the $(j+1)^{th}$ neuron of this layer instead of the first one. Consider any pair
334 of polytopes $\mathcal{P}_A, \mathcal{P}_B \in \mathscr{P}_{k,j}$ with nonempty intersection. These must be PC, and in particular
335 their union must either be fully contained in some $\mathcal{P}_C \in \mathscr{P}_{k,0}$ or not. If so, then there exists some
336 hyperplane $H_C$ such that $G_{j+1} \cap \mathcal{P}_C = H_C \cap \mathcal{P}_C$ and thus $\mathcal{P}_A \cap \mathcal{P}_B \cap G_i = \mathcal{P}_A \cap \mathcal{P}_B \cap H_C$ so we
337 satisfy the criteria to apply lemma 3.4. If there is no such $\mathcal{P}_C$, then $\mathcal{P}_A \cap \mathcal{P}_B$ must be a facet of each
338 of them, $F$. Then we can mimic the argument in the previous paragraph to show that the polytopes
339 generated by the addition of $G_{j+1}$ remain PC.

340 Finally, we need to prove that the decision region of $F$ at $x_0$ forms a polyhedral complex. Let $\mathscr{Q}$
341 be the collection of linear regions of $F$ that have a nonempty intersection with the decision region
342 of $F$ at $x_0$. As any subset of a polyhedral complex is also a polyhedral complex, $\mathscr{Q}$ is certainly a
343 polyhedral complex. Let $F(x_0) = i$ and let $g_j = \{x | f_i(x) \geq f_j(x)\}$. For each linear region of $f$, $g_j$
344 is a halfspace. The decision region of $F$ at $x_0$ is exactly $\{\mathcal{Q}_i \cap (\bigcap_{j \neq i} g_j \mid \mathcal{Q}_i \in \mathscr{Q}\}$. It suffices to
345 show that for a single $j$, $\{\mathcal{Q}_i \cap g_j(x)) \mid \mathcal{Q}_i \in \mathscr{Q}\}$ is still a polyhedral complex, as we can iterate over
346 all $j \neq i$. Then for a fixed $j$ and any $\mathcal{Q}_i, \mathcal{Q}_k \in \mathscr{Q}$ with nonempty intersection, and letting $g_j(\mathcal{P})$ be
347 the hyperplane defining $g_j(x)$ for the linear region $\mathcal{P}$, we note that $\mathcal{P} \cap \mathcal{Q} \cap g_j(\mathcal{P}) = \mathcal{P} \cap \mathcal{Q} \cap g_j(\mathcal{Q})$.
348 This is exactly the criteria required to apply lemma 3.4, which maintains that the pair of polytopes $\mathcal{P}$
349 and $\mathcal{Q}$ lying in the decision region are PC. This holds for every pair of polytopes in $\mathscr{Q}$ with nonempty

350 intersection, so $\mathscr{Q} \cap g_j$ is a polyhedral complex, and hence so is the entire decision region of $F$ at
351 $x_0$. □

352 In fact, the following corollary demonstrates that except in extreme cases, the facets of each linear
353 region correspond to exactly one neuron flipping configurations.

354 **Corollary 4.3.** *If the network parameters are in general position and $A, B$ are neuron configurations*
355 *such that $dim(\mathcal{P}_A) = dim(\mathcal{P}_B) = n$ and their intersection is of dimension $(n-1)$, then $A, B$ have*
356 *hamming distance 1 and their intersection corresponds to exactly one ReLU flipping signs.*

357 *Proof.* As both $\mathcal{P}_A$ and $\mathcal{P}_B$ are of full dimension, no coordinate of the neuron configurations $A, B$
358 can be zero. Under the assumption of general position of the network parameters, the halfspace
359 that defines each polytope constraint lies in a different $(n-1)$-dimensional affine subspace, hence
360 each facet corresponds to exactly one neuron. Indeed, each facet of each linear region's polytope
361 corresponds to at exactly one ReLU constraint being set to equality. Since $dim(\mathcal{P}_A \cap \mathcal{P}_B) = n - 1$
362 and since $\mathcal{P}_A, \mathcal{P}_B$ are PC, $\mathcal{P}_A, \mathcal{P}_B$ must be a facet of each of them. This facet is a linear region of
363 the network as well, corresponding to a neuron configuration $C$ that is identical to $A, B$, but with
364 some coordinate set to zero. As $A \neq B$, and the neuron configuration $C$ has exactly one zero, it must
365 be the case that the hamming distance between $A$ and $B$ is exactly one, corresponding to exactly one
366 ReLU flipping signs. □

# F  An Approach For Computing Tighter Upper Bounds

368 As mentioned in Section 5, maintaining a nontrivial upper bound on the pointwise robustness
369 accelerates the runtime of GeoCert by restricting the domain we have to search. This has a twofold
370 benefit as (i) this allows us to quickly reject potential facets as infeasible by checking if their
371 containing hyperplane intersects the restricted domain, and (ii) allows for tighter pre-ReLU activation
372 bounds to be computed. This latter point allows for potential facets to be rejected without the
373 computation of their projection as Corollary 4.3 implies that neurons that are stable within a domain
374 do not correspond to any facets inside that domain.

375 Fortunately, there has been an explosion in the field of computing upper bounds to the pointwise
376 robustness, typically described as adversarial examples. In this section we present a variant of the
377 attack techniques presented in [3, 7, 4, 1]. Our goal is to be able to compute a reasonably tight upper
378 bound for a single example in a very short amount of time. In general, attack techniques are viewed
379 as optimizations over some perturbation that aims to maximize a loss that is large when the classifier
380 makes a mistake. We discuss two popular existing adversarial attacks from an .

381 One attack, known as PGD performs *gradient ascent* directly on the loss ands projects at each iteration
382 back onto a set of allowable perturbations. Letting the allowable set of perturbations be $B_p^\epsilon(0)$ and
383 the domain of valid images be $\mathcal{D}$, then the allowable set of adversarial perturbations for image $x_0$ is
384 $\mathcal{D}' := B_p^\epsilon(0) \cap \{x - x_0 \mid x \in \mathcal{D}\}$. PGD seeks to solve the maximization problem

$$\max_{\delta \in \mathcal{D}'} \mathcal{L}(x_0 + \delta, y) \tag{22}$$

385 where $\mathcal{L}(\cdot, y)$ is some loss that is small when the network classifies its argument as class $y$, and large
386 otherwise. The PGD iterations become

$$\delta^+ = \Pi_{\mathcal{D}'}\left(\delta + \eta \nabla_\delta \mathcal{L}(x_0 + \delta, y)\right) \tag{23}$$

387 Notice that the goal of PGD is not to induce a minimal distortion adversarial example, but simply to
388 minimize classifier accuracy within a fixed threat model. We also note several tricks that are useful
389 in practice such as a random initialization of $\delta \in \mathcal{D}'$ and repeated restarts to find more successful
390 adversarial examples.

391 An alternative attack, pioneered by Carlini and Wagner [1] does aim to produce low-distortion
392 adversarial examples by simply letting $\mathcal{D}' := \{x - x_0 \mid x \in \mathcal{D}\}$ and solving the optimization

$$\min_{\delta \in \mathcal{D}'} \quad ||\delta|| \tag{24}$$

$$\text{s.t.} F(x_0 + \delta) \neq F(x_0)$$

$$\tag{25}$$

**Input** classifier $f$, input $x_0$, initSize $\nu$, ballSize $\epsilon$
        lr $\eta$, numIter $n$, numRand $r$
        numBin $k$
  **for** $i \in [r]$ **do**
    $u_i = \infty$
    $\delta_i \leftarrow RandBall(\nu)$
    **for** $iter \in [numIter]$ **do**
      $\delta_i \leftarrow \Pi_\epsilon(\delta_i + \eta \nabla f(x + \delta_i))$
    **end for**
    **if** $f(x + \delta_i) \neq f(x)$ **then**
      $\delta_i \leftarrow BinSearch(f, x_0, \delta_i, k)$
      $u_i \leftarrow ||\delta_i||_p$
    **end if**
  **end for**
**RETURN** $\min_i u_i$

**Algorithm 1:** Fast Upper Bound

**Input** classifier $f$, point $x_0$
        perturbation $\delta$, numIter $n$
  $lo \leftarrow 0, \ hi \leftarrow 1$
  **for** $i \in [n]$ **do**
    **if** $f(x_0 + (lo + hi)/2 \cdot \delta) \neq f(x_0)$ **then**
      $hi \leftarrow (lo + hi)/2$
    **else**
      $lo \leftarrow (lo + hi)/2$
    **end if**
  **end for**
**RETURN** $hi \cdot \delta$

**Algorithm 2:** BinSearch

Where the adversarial constraint is typically put into the lagrangified form with the best multiplier found via binary search:

$$\min_{\delta \in \mathcal{D}'} \quad ||\delta|| + \lambda G(x_0 + \delta) \tag{26}$$

Where $G$ is a function that is zero everywhere where the classifier makes a mistake, and positive elsewhere. This is then solved with a standard gradient descent algorithm. The main critique of this method is that the binary search over the hyperparameter $\lambda$ dictates the runtime be several times longer than PGD. Note that during this optimization, once the intermediate iterate is outside $x_0$'s decision region, the gradient steps push the intermediate iterate radially inwards. However, unless step sizes are tuned nicely, many iterations with the radially-inward direction may be taken.

We provide a tweak to PGD that allows one to quickly generate adversarial examples that are optimized to have minimal distortion. This technique is as follows: for example image $x_0$, compute many random perturbations on $x_0$, and run PGD with a large domain on each of these randomly perturbed starting points. Once complete, collect each of the examples for which the classifier makes a mistake. Run a binary search along the line connecting the example and the starting point $x_0$, in an attempt to 'project' onto the decision boundary. Return the minimal-distance of these projected adversarial attacks as the adversarial example for $x_0$.

The binary search step requires only forward passes and is significantly faster than the several gradient steps required by CW to 'project' back to the decision boundary. This allows one to effectively perform a quick PGD attack, which is almost always successful under a sufficiently large threat model, but also attain a successful adversarial attack with small distortion.

We note, the emphasis here is not on attaining the minimal distortion adversarial example, but on speed and guaranteed success. Our goal is to very quickly find an adversarial example that is incentivized to be close to the original point and will almost always succeed.

Figure 4: evidence that verification for trained nets does not follow worst case behavior

## G   Extra Experiments

### G.1   Extra Experiment 1:

To reiterate, in the worst case our algorithm may need to explore an exponential number of polytopes. Here, we provide results which seem to suggest that for PLNNs trained on MNIST the number of polytopes is well removed from the worst case. Figure 4 shows the number of polytopes encountered in an $\ell_\infty$ ball of size $t$ around several random images. (Note that the relevant network in this case is the 70NetBin network described previously.) The distance $t$ is increased until the region around each of the sampled points includes the entire domain for MNIST (i.e. [0, 1] hypercube). Thus, the maximum number of polytopes that could be encountered for this problem is very loosely upper bounded by 73. On average, the number of polytopes encountered for this example would be closer to 6 as the average distance is 0.19. This plot seems to suggest that the number of polytopes encountered is much smaller than the worst case possibility.

### G.2   Extra Experiment 2:

Additionally, we run experiments to investigate the benefit of using a Lipschitz overapproximation based potential versus the standard $\ell_p$ distance. Table G.2 demonstrates the average number of encountered polytopes when verifying pointwise robustness.

Table 1: Average number of polytopes explored until computing exact pointwise robustness across binary (1's and 7's only) MNIST, and full MNIST, and two architectures. The average is over 50 random examples. This demonstrates the benefit of leveraging the Lipschitz upper bound in the potential function.

|           | Binary MNIST | | Full MNIST | |
|-----------|-------|-------|-------|-------|
| Potential | 70Net | 40Net | 70Net | 40Net |
| $\phi_{lip}$ | 4.2 | 15.3 | 9.7 | 27.5 |
| $\phi_p$ | 5.1 | 25.6 | 17.1 | 90.3 |