[Reviews · NeurIPS 2019]

Reviewer 1



The paper is interesting and well-written. It is a little dense, which is somewhat unavoidable given the page limit. It may help if the part on "Graph Theoretic Formulation" was expanded in some way since the high level description is a bit difficult to follow. To make space, it might be possible to move the mention of some of the lemmas to the supplementary material. I did not go through the proofs in detail.

Reviewer 2



1. Originality: High. I think this is quite different from most previous work that involves propagating bounds through the network (for incomplete verification), and MLP/SMT based exhaustive search approaches. 2. Quality: The quality of the work is quite high. My biggest concern is with the experiments (see section on suggested improvements). 3. Clarity: Good. The paper is largely very well written, and quite easy to follow. I appreciate the effort the authors put into the manuscript. 4. Significance: Low. While the approach is new, the evidence is not strong enough here to believe that this is indeed a tractable approach for verifying the exact robustness for deep networks.

Reviewer 3



The main contribution of this work is a new approach to neural network robustness based on polyhedral complexes. Quality: The theoretical results are clear and intuition explained. (I haven't verified the proofs in the appendices). The proposed algorithms are clever and neatly explained. The main missing component is more extensive empirical evaluation: -- 128 random samples seems very small to draw reliable conclusions -- Only one type of network is considered (binary classification 1 vs. 7 on MNIST via standard training): The performance of a verifier typically varies a lot depending on how the network was trained (whether it was trained to be robust and by which method). For example, training to enforce relu stability would reduce the iteration time as mentioned in the paper. -- The main gain of the proposed approach is under limited time constraints, it can verify much more than complete methods like MILP. How does this compare to other LP/SDP based incomplete verifiers? Originality: The approach towards certifying robustness of neural networks is novel and very different from existing approaches. However, I am not familiar with the geometric results proposed to comment on whether the theorems involve novel techniques over existing work in the area. Clarity: The main ideas of the paper are well presented and the intuition is clearly explained. The technical exposition could be improved in parts, especially in defining and using terms that are not standard in ML literature. Some minor specific points: a) Corollary 3.2: T refers to boundary of the polytope but isn't mentioned in the statement of the corollary which is quite confusing on first read. b) The definition of convex polytope in Section 3 seems to be missing convexity requirements c) Chebyshev ball isn't formally clearly d) A figure to represent the bipartite graph could aid understanding Significance: i think this work is significant as it provides a new approach to quantifying robustness of a network. While the preliminary results presented in the paper do not yet address scalability issues, and provide only a small improvement over the baselines considered, this new approach could propel innovations for certifying neural network properties based on similar geometric ideas. -- Author feedback response -- I thank the authors for their response and clarification.

[Author Response · NeurIPS 2019]

First we thank the reviewers for their helpful comments and insightful questions. We have addressed typos and expanded definitions for terms that are nonstandard in machine learning literature. Reviewer #1 primarily commented on the density of the paper and the clarity of certain technical components of our approach. Reviewers #3 and #4 are interested in a more comprehensive experimental section. Primarily, reviewer #3 raised several questions about the methodology by which we compared against the mixed-integer programming (MIP) approach. Reviewer #4 raised questions about the comparisons against leading state-of-the-art incomplete verifiers. Both reviewers also were interested in experiments comparing our technique against MIP for networks trained in different ways (e.g. adversarial training, or training with a loss function that regularizes towards ReLU stability or linear-region maximization).

**Reviewer #1:**

**"It may help if the part on "Graph Theoretic Formulation" was expanded in some way since the high level description is a bit difficult to follow."** We have shifted some of the lemma statements to the appendix in order to clarify the "Graph Theoretic Formulation" and include an illustrative figure.

**Reviewer #3:**

**"The paper presents a scenario where the algorithm does better than the MIP solver from Tjeng et al., in the case where we enforce a time-out. I am not convinced that this is the right metric to evaluate the baseline on..."** We do concede that our empirical results demonstrate that for very small networks – for which computing pointwise robustness is tractable – the MIP approach terminates more quickly than ours. However, the hardness of even approximating pointwise robustness, coupled with the massive overparameterization of current state-of-the-art networks implies that termination for complete verification is a daunting task. Accepting that termination may be intractable, this spawns two possible alternative tasks for robustness verification: (i) "Can a certain region be certified as safe?" or (ii) "What is the largest certifiably safe region that can be found under a fixed time limit?". Fundamentally, MIP/SMT approaches are formulated in a way to answer question (i) – a property also shared by most incomplete verifiers. However the choice of adversarial region in the literature is often 'arbitrary' and it makes sense to consider techniques geared towards solving formulation (ii). In that respect, and as we demonstrate empirically, MIP solvers falter because they are *non-local* and operate by recursively tightening relaxations to integral constraints, which correspond to ReLU configurations.

Regarding improvements towards the binary search schedule for MIP: while there might be clever heuristics that can be applied to optimize this search schedule, our attempts at finding better schedules did not drastically alter our results. We find that the MIP, once asked to verify a region for which the convex relaxations are not infeasible, branches extensively to explore many nodes and effectively 'hangs' and provides no further useful information. Without exploring this too much further, we felt a fair comparison was to rely on the schedule provided by Tjeng et. al. Starting at the bound provided by Fast-Lip (or similar) is a nice idea, but upon incorporating this, we find this only provides improvements up to the bound Fast-Lip provides.

**"It would also help to include an ablation study comparing the effect of the various heuristics for the problem. Also, would some heuristic based training (such as https://arxip.org/abs/1810.07481) help your approach be more scalable?"** We will include this ablation study in the final version. In general, the bound propagation techniques and heuristics applied at evaluation time were applied to both methods. To improve internal development speed we typically applied all heuristics provided in Tjeng et. al to both methods, in addition to the one we develop in Appendix F. We do believe training techniques such as ReLU stability or maximization of linear regions would help both our approach and MIP to be more scalable. This is an interesting area for future research, particularly when using GeoCert as a tool to explore properties of the linear regions of such networks (see, for example, appendix G.2 where we leverage GeoCert to *exactly count* the number of linear regions in the image domain).

**"For the wins you report (with early stopping), does this hold for larger networks as well?"**: Yes. For larger networks, particularly in the $\ell_2$ domain, this is even more pronounced. We showed the case for early stopping with a timeout of 300 seconds, but this result holds for larger networks with similar timeout parameters and another table will be included in the final version.

**Reviewer #4:**

**"128 random samples seems very small to draw reliable conclusions."** Our results hold as we increase the number of samples. We will include these numbers in future revisions.

**"How does this compare to other LP/SDP based incomplete verifiers?"** While incomplete verifiers which answer the 'decision problem' formulation (such as LP/SDP approaches) may provide marginally better starting points than Fast-Lip after a binary search, these approaches are fundamentally limited and will not asymptotically become tight. Also note that tighter Lipschitz estimation techniques (such as RecurJac) can be incorporated into GeoCert and provide a better starting point.

[Meta-Review · NeurIPS 2019]

Thank you for your submission to NeurIPS. Although there is still some disagreement amongst the reviewers (specifically on how well the proposed method compares to GeoCert in practice), the general consensus is also that the proposed approach presents an interesting and valuable contribution to the class of exact verification methods for deep networks. Some additional discussion to competing approaches, mentioned the rebuttal, would help to strengthen the paper further, but overall the majority feeling is that this is a very nice contribution to the topic.